# The Role of Talking Faces in Infant Language Learning: Mind the Gap between Screen-Based Settings and Real-Life Communicative Interactions

**DOI:** 10.3390/brainsci13081167

**Published:** 2023-08-05

**Authors:** Joan Birulés, Louise Goupil, Jérémie Josse, Mathilde Fort

**Affiliations:** 1Laboratoire de Psychologie et NeuroCognition, CNRS UMR 5105, Université Grenoble Alpes, 38058 Grenoble, France; louise.goupil@univ-grenoble-alpes.fr (L.G.); jeremie.josse@univ-grenoble-alpes.fr (J.J.); mathilde.fort@univ-grenoble-alpes.fr (M.F.); 2Centre de Recherche en Neurosciences de Lyon, INSERM U1028-CNRS UMR 5292, Université Lyon 1, 69500 Bron, France

**Keywords:** talking faces, audiovisual speech perception, infancy, language acquisition, naturalistic interactions, audiovisual speech cues

## Abstract

Over the last few decades, developmental (psycho) linguists have demonstrated that perceiving talking faces audio-visually is important for early language acquisition. Using mostly well-controlled and screen-based laboratory approaches, this line of research has shown that paying attention to talking faces is likely to be one of the powerful strategies infants use to learn their native(s) language(s). In this review, we combine evidence from these screen-based studies with another line of research that has studied how infants learn novel words and deploy their visual attention during naturalistic play. In our view, this is an important step toward developing an integrated account of how infants effectively extract audiovisual information from talkers’ faces during early language learning. We identify three factors that have been understudied so far, despite the fact that they are likely to have an important impact on how infants deploy their attention (or not) toward talking faces during social interactions: social contingency, speaker characteristics, and task- dependencies. Last, we propose ideas to address these issues in future research, with the aim of reducing the existing knowledge gap between current experimental studies and the many ways infants can and do effectively rely upon the audiovisual information extracted from talking faces in their real-life language environment.

## 1. Introduction

Speech is typically perceived multimodally: most social interactions occur face to face, so we not only hear our interlocutor but also see the information inherent to any facial articulatory movement and inherent to linguistic communication more specifically. Primarily, through the eye’s region, faces convey a good amount of information about a person’s state of mind, attitude, and potential intentions, as well as referential information and speech rhythms (see [1] for a review). Additionally, through the talker’s mouth region, we gain access to spatiotemporal and acoustic congruent auditory and visual speech cues [2,3]. Given that auditory and visual cues provide overlapping information about the same speech event, they have often been named “redundant audiovisual (AV) speech cues” (e.g., [4,5]). Prior studies have shown that access to such AV redundant speech cues, compared to auditory-only situations, can facilitate lexical access and, more largely, speech comprehension, most notably when the acoustic signal becomes difficult to understand due to noise [6,7,8,9,10,11] or to an unfamiliar accent or language (e.g., [12,13,14]). In such occasions, adult listeners have been shown to increase their visual attention (hereafter attention) to the talker’s mouth in order to maximize the processing of AV speech cues and enhance their processing of speech; for instance, when background acoustic noise increases [15,16], when volume is low [17], when their language proficiency is low [18,19,20], or when they are performing particularly challenging speech-processing tasks (e.g., speech segmentation [21] or sentences comparison [18]). On the other hand, when speech-processing demands are reduced, adults modulate their attention and focus more on the eyes of the talker [21,22,23], which can also support language understanding by constraining interpretations (e.g., of the speaker’s current emotion or current focus of attention).

Overall, these studies suggest that adults flexibly modulate their selective attention to subparts of a talking face as a function of factors such as the task at hand, the amount of noise in the sensory environment, and language experience. If adults do this to maximize the processing of redundant AV speech cues inherent in talkers’ faces and help them process speech, we might hypothesize that infants could also take advantage of such cues when learning their first language/s.

In the present article, we review existing evidence on whether and how the infant’s visual perception of dynamic talking faces is linked to language learning. First, we do so by describing findings from prior screen-based studies on infant face perception, including selective attention to talking faces and its relation to language learning (Section 1). Second, we present evidence from naturalistic studies on infants’ perception and attention to their caregiver’s faces during play and discuss the apparent differences in looking patterns towards talking faces that have been observed between screen-based and real-life studies. Third, we describe three variables (i.e., social contingency, speaker-dependency, and task-dependency) that we believe are currently understudied and should be further explored to better understand how infants dynamically and flexibly exploit talking faces to learn their native language/s (Section 3). Finally, we present our conclusions and directions for future studies (Section 4) and propose several ideas to reach a more complete knowledge of the role that speakers’ talking faces play in infants’ language acquisition.

## 2. The Role of Faces in Infants’ Attention and Language Acquisition: Evidence from Screen-Based Settings

### 2.1. Infants’ Preference for Faces

During their first year of life, most infants’ direct communicative interactions occur face to face with their caregivers (for an estimation in ecological situations, see [24,25]), which allows them to gain access to rich AV cues that dynamic talking faces provide. Crucially, on top of the fact that faces are readily accessible in infants’ typical visual field, screen-based studies attempting to mimic these face-to-face situations in the lab have revealed that infants also show a preference for face-like stimuli from very early on: they tend to orient toward faces preferably when in competition with other stimuli (e.g., [26]). It has been suggested that this early “face bias” is already present in newborns, who look longer on face-like patterns than inverted face-like patterns or different geometric patterns [27,28,29], although it is unclear whether this reflects a genuine preference for “faces” per se rather than a preference for some lower-level visual features of faces. Later, once infants’ visual system begins to be fully functional around 3 to 4 postnatal months [30], and they have gained some experience with their environment, infants’ preference for faces continues to increase, particularly when presented amongst multiple competing objects (4- to 12-month-old infants [31,32]). These studies suggest that infants routinely have a direct visual access to talking faces, and that they show a very early interest and increased attention to such information. Other studies suggest that infants’ attentional preference for faces after the age of 3–4 months cannot entirely be explained by low-level visual features, reflecting the onset of an active search for relevant information in faces [31]. In line with this hypothesis, studies have also shown that infants do not attend to speakers’ faces as much when they do not engage in communicatively relevant interactions, such as a mutual gaze [33,34], which enhances their cortical processing of faces and face recognition skills [35,36].

In summary, it has been discussed that learning to process interlocutors’ faces is likely to be a crucial aspect of infants’ social [37,38] and language learning (for recent reviews, see: [39,40,41]). However, which specific facial cues (i.e., eyes and eye-gaze, AV speech cues from the talker’s mouth) infants attend to and exploit at each moment in development and for which type of learning they might do so remains to be understood.

### 2.2. Infant’ Attention to the Eyes and Mouth of a Talking Face, and Its Relation to Language Acquisition

Prior studies have approached this question by exploring infants’ selective attention to videos of talking faces, analyzing their looking time to the internal features of a talker’s face (i.e., eyes and mouth) across development. Furthermore, some have examined how such measures of attention relate to infants’ concurrent and later language outcomes, providing correlational evidence that attending to the eyes or the mouth region of a talking face at different stages of development can support language learning (e.g., [42,43,44,45,46,47,48]).

On the one hand, studies suggest that paying attention to the eye region of a face can boost infant language learning: a direct gaze—rather than an indirect gaze—establishes the intent to communicate, and young infants are sensitive to these communicative signals [33,34,49,50]. Later, infants develop gaze- and head-following skills, which allow them to orient their attention at the same location as their social partners, creating joint attention moments between the infant and the caregiver, which can support word learning (see [41,51] for recent overviews). Of particular relevance for lexical acquisition, following the gaze/head of a speaker in the direction of a named object can allow listeners to disambiguate which object is associated with the novel spoken label [52], thus solving the referential uncertainty problem (i.e., the uncertainty inherent in mapping a heard name to its intended referent, in a complex and variable environment [53]). In line with this idea, several studies have shown that gaze-following behavior in the first year is positively correlated with vocabulary development in the second year [43,54,55,56]. Additional experimental evidence suggests that infants’ encoding of the visual property of a novel object can be increased when they look toward it by following another person’s gaze [35]. Moreover, gaze following also supports infants’ mapping of novel words to the object that the speaker is looking at [41,57].

On the other hand, paying attention to the mouth of a talking face can also be useful for enhancing speech processing and infant language acquisition (e.g., [58]). As reviewed above, adults rely on the AV speech cues of a talker’s mouth to enhance speech perception, especially when speech processing becomes challenging (e.g., [15]). Prior evidence shows that infants are sensitive to AV correspondences from 2 to 3 months of age (e.g., [59,60]). Their auditory perception of speech is influenced by its concurrent visual presentation from 5 months of age (i.e., McGurk effect; [61]), and they can use AV speech information to discriminate and learn speech sounds [62,63], segment speech [64] and increase word processing and retention [65,66]; see for reviews [39,41]. It is then reasonable to hypothesize that orienting and sustaining one’s attention on a talker’s mouth could be a good strategy for enhancing language acquisition at various levels during infancy.

Lewkowicz and Hansen-Tift [42] performed the first study to explore the relationship between infants’ selective attention to a video of a talker’s face (i.e., to the eyes and mouth) across their first year of life and its relation to language processing [42]. To do so, the authors recorded the gaze of 4- to 12-month-old monolingual English-learning infants whilst watching a speaker talk in their native language. The results first showed that similar to newborns and younger infants [26,67], 4-month-old infants attended more to the talker’s eyes. Crucially, however, infants then started shifting their attention toward the mouth of the talking face, showing equal attention to the eyes and mouth at 6 months and more attention to the mouth from 8 months of age [42].

This attentional shift from the eyes to the mouth of the talker was interpreted by the authors as evidence of infants’ emerging interest in processing AV speech. Given that 6 months of age is also the stage where infants’ endogenous control starts to emerge [68,69], this would allow them to exert top-down control to specific subparts of a speaker’s face and voluntarily deploy more of their attention into the speaker’s mouth. Indeed, the onset of this mouth preference has been confirmed to emerge at around 6–8 months of age by more recent studies [44,45,46,47,70,71,72,73] and seems to present its peak at around 18 months of age [46,74]. This disposition to look toward speakers’ mouths remains present in the second year [75,76] and slowly diminishes during later childhood, with 5-year-old children typically showing balanced attention between the talker’s eyes and mouth when perceiving native speech [46,77,78].

Overall, this suggests that infants and children deploy an important amount of their selective attention to the mouth of talking faces during their first years of life and that such preference for the mouth diminishes with increasing age and language proficiency. However, to what extent are infants’ attentional patterns to the subparts of talking faces reflective of their interest in the processing of AV speech cues (rather than being determined exogenously as a function of relative saliency)? Additionally, is this attentional strategy efficient in helping them learn their native language/s?

Recent studies have provided empirical evidence supporting the link between attention to the mouth and different aspects of language processing. For instance, studies have shown that infant-directed speech (IDS)—which is known to enhance infant attention to speech, but also language processing and learning [79]—elicits greater attention to the mouth during the first and second year of life [75,80,81]. The fact that IDS induces infants to attend more to the mouth of talkers suggests that this may be an efficient strategy of IDS to increase the processing of AV speech cues and, in turn, boost infants’ processing of speech. In a similar vein, a recent study has shown that 15-month-old infants also increased their attention to a talker’s mouth as they learned new non-adjacent rules from unfamiliar speech [82]. This has been interpreted as reflecting infants’ additional reliance on AV speech cues to help process and acquire the syntactic rules of their native language.

Additional evidence comes from studies showing that similar to adults, infants also increase their attention to the mouth of a talker when speech processing becomes more challenging, as is the case of perceiving non-native speech [42,47,76,83] (although not in [46]) or growing up in bilingual environments and learning two rhythmically and phonologically close languages, such as Spanish and Catalan. In the latter case, close-language bilingual infants deploy more of their attention to a talker’s mouth than monolingual infants or bilingual infants learning phonologically distant languages [47,76,84,85], which continues into later childhood [76,86] and is likely to enhance language differentiation and processing under a linguistically more challenging environment. Taken together, these findings suggest that, like children and adults, infants already adapt their looking patterns to subparts of a talking face flexibly and deploy more or less attention to the AV speech cues of a talker’s mouth as a function of local demands concerning speech processing and language learning.

Crucially, some studies have also provided direct correlational evidence showing an association between infants’ increased attention to the mouth and the different aspects of language learning. For instance, Young and colleagues [73] recorded 6-month-olds’ eye gaze in a video of a live mother–infant interaction and revealed that greater mouth-looking predicted higher expressive language at 18 months. In the same vein, Tenenbaum and colleagues [43] showed that looking time at a talker’s mouth as well as gaze-following behavior at 12 months of age predicted infants’ vocabulary size at 18 and 24 months of age. Using a different paradigm, Imafuku and colleagues [48] presented 6-month-old infants with videos of a speaker producing vowel sounds, while recording their eye gaze and vocal productions. The results showed that infants vocalized more when their faces were upright, made eye contact, and when infants looked more at the speakers’ mouths [48]. The correlation between mouth-looking and language skills has also been shown by other studies, albeit with differences in age and the specific aspects of lexical development (see [44,45]).

To summarize, it emerges from these findings that, over time, infants increasingly deploy their attention toward the AV speech cues of a talker’s eyes and mouth regions to help them process and learn their native language/s. However, it is worth noting that these conclusions originate from screen-based studies. Such experimental situations generally differ from real-world interactions (for a recent overview, see: [87]) and, therefore, we must also explore whether and how these mechanisms are effectively used and relied upon by infants in their complex and culturally situated language environment, and during free-flowing, dynamic social interactions like proto-conversations, play, etc. In the following section, we thus explore infants’ selective attention to caregivers’ faces in real-life language learning situations.

## 3. The Role of Faces in Infants’ Attention and Language Acquisition: Evidence from Real-Life Communicative Interactions

During real-life interactions, talking faces appear within the realm of complex, multimodal, and only partially predictable socio-communicative interactions. This is a different situation compared to what has happened in most studies examining the role of talking faces for infant language learning using screen-based tasks. Indeed, in virtually all these studies (including some work from the authors), non-contingent face stimuli have often appeared exaggerated in size and presented against a rather neutral background with no or very few distractors on a computer screen (e.g., [42,44,46,70,72,74,76,84]). These presentation characteristics could lead to a distortion in the perceptual saliency of talking faces and their role in guiding infants’ attention and language processing and learning.

Bahrick and colleagues posited that redundant and temporally synchronized AV cues, such as the ones provided by the talking mouths of speakers, “pop out” from the background and other less salient distractors, directing attentional selectivity (i.e., the Intersensory Redundancy Hypothesis, IRH, [88,89]). In other words, they suggest that talking faces offer a form of pre-attentive bottom-up saliency that would automatically attract the perceiver’s visual attention when surrounded by other less salient distractors. This idea has been previously challenged, however, by studies in adults showing that the detection of temporally synchronized and coherent AV cues is not a pre-attentive process given that it is sensitive to attentional demands [90,91,92] and that it requires a top-down serial time search, both in adults [93] and in children [94].

Another prediction that derives from IRH is that talking faces should be highly and frequently looked at when present in an infant’s environment. This preference for talking faces should be observed from around 2 to 3 months of age once infants become sensitive to AV correspondences [59,60]. At this developmental stage, AV correspondences should make faces more salient and catch infants’ attention in a bottom-up fashion. As cognitive control matures with age, the preference for talking faces should increase, caused by both the bottom-up system, responding to the highly salient features of AV speech, and by gradually developing top-down forms of endogenous attention [68,95], assuming that infants endogenously seek the social and linguistic information that talking faces provide.

Screen-based studies support this hypothesis by showing that infants prefer to look at silent static or talking faces rather than objects in their first months of life [28,96,97] and that crucially, this preference increases with age in typically developing infants [31,32,98,99] (although see [97] for an age decrease) independent of the low-level saliency features of faces [31,98]. In such screen-based studies, when exploring the role of talking faces in early language acquisition, the estimated percentage of looking time to faces has been reported between 30 and 90%, depending on age and stimulus features [31,32,98,99,100].

Differently, however, if we extract descriptive statistics from real-life studies that examine how infants visually explore their environment in naturalistic contexts, which are thought to afford language learning (e.g., joint play), data suggest that faces comprise a smaller percentage of infants’ visual input (i.e., between 10 and 30%), and that such a percentage seems to decrease with age [24,25]. For instance, in free toy play situations, infants from 11 to 12 months of age achieve joint attention moments with their caregivers toward objects that are close to them (e.g., [101]) with little attentional guidance from their caregiver’s faces and head direction [102,103,104,105], and a seemingly greater impact of manual gestures [106]. These data could suggest that faces play a minor role in children’s social and language learning, at least after 11–12 months of life. It is expected that in free-flowing interactions, with higher stimulation and competition, infants should show lower face-looking times than in screen-based studies. Yet, this should remain constant over time; therefore, the fact that this preference decreases with age remains to be explained, as it contradicts the increase in age shown in screen-based studies. In other words, collectively, these findings suggest that the perceptual saliency of faces in naturalistic settings might be lower than what we would predict based on screen-based studies and the IRH [89].

Relatedly, infants’ free vs. restrained movement reasonably have a strong impact on infant attention. As infants grow up, motor and postural developments progressively transform their visual input, increasing its complexity and diversity [107,108] and affecting the relative saliency of talking faces. Infants shift from laying on their back and mainly looking up—towards the ceiling and the face(s) of their caregiver(s)—to sitting, then standing up and walking. Such novel postural and motor control affords infants a novel perspective of their environment by looking more frontally, reaching and grasping objects that surround them. Under this rationale, directly looking toward adults’ faces becomes less frequent, and infants engage more with specific subsets of their visual environment (e.g., hands, objects, food, etc., see [108]).

Overall, these studies exploring naturalistic interactions where infants move freely suggest that faces may be perceptually less salient and less frequently attended to than previously assumed based on screen-based studies and that, with increasing age, infants’ attention to faces diminishes. This suggests that infants’ need to deploy their attention toward the AV speech cues of speakers’ faces may also gradually decrease with increasing age and language proficiency. A potential explanation is that, indeed, infants’ information-seeking behavior becomes more and more optimal as they grow up and that, therefore, given that their cognitive and linguistic abilities improve, their selective attention to faces becomes less frequent but more targeted to certain specific situations when AV speech cues become relevant. These may involve situations where communicative information becomes unclear (due to referential uncertainty, speech ambiguity, noise, L2 speech, etc.) or when other social and emotional factors come into play (face identification, emotional reassurance, etc.). Therefore, infants may prioritize other stimuli while communication is clear and fluent and only look toward speakers’ faces in ambiguous situations.

In our view, these discrepancies might also arise from the fact that three factors have been insufficiently considered in past studies, even though they systematically vary by context and might deeply impact the results that are observed in any given context. In the following section, we provide the reader with an overview of these three factors that we believe may influence infants’ attention and the use of talking faces in naturalistic settings, which are currently understudied. In turn, we also discuss how these factors may also help explain the differences in attention to faces between the screen-based and real-life settings above described.

## 4. Moving Closer to Real-Life Language Learning Interactions: Three Factors That Deserve More Attention

### 4.1. The Influence of Dyadic Bidirectional Contingency

The first factor that we consider to be understudied in screen-based studies exploring the role of talking faces in language learning—that has also not been systematically explored or manipulated in naturalistic studies—concerns the bidirectional contingency between the infant and the talking face the infant is looking at. Typically, screen-based approaches provide very little temporal contingency between the shown stimuli and the infant’s behavior and mainly focus on the unidirectional adaptation of the infant to talking faces rather than considering this relation in a bidirectional fashion (e.g., [42,46,48,76]). In other words, in most screen-based paradigms, the behavior of the talking face is determined by the experimenter (e.g., pre-recorded) rather than by the participant. Talking faces almost never adapt to infants’ attentional variations across time (e.g., gaze direction), including emotional or sociolinguistic responses (e.g., smile, pointing, presence of babbling, or imitation of the speaker). Overall, this lack of bidirectional contingency could lead to an underestimation of the dynamic components of dyadic interactions in real-life situations, which are known to modulate infants’ attention and language learning, e.g., [101,109,110,111]. More specifically, it could lead to an overestimation of the infants’ adaptation to the caregiver’s face behavior and/or underestimating the adaptation of the caregiver’s face as a response to the infant’s behavior.

Studies examining real-life interactions between infants and their parents (e.g., during joint play) have shown large inter-individual variability in the extent to which caregivers tend to contingently respond to their infants’ behaviors (a measure referred to as contingency, sensitivity, or responsiveness), and this seems to be a unique and important predictor for infant word learning [112,113,114]. For instance, the more caregivers show temporally contingent responsivity to their infant’s behavior, the more their infant is attentive [115,116,117], learns novel rules [118], novel words [112,113,114,119], remains sensitive to non-native phonetic contrasts [120,121,122], and produces more mature speech-like vocalizations [111]. The importance of social contingency for learning has been suggested to stem from the fact that it allows infants to better predict the consequences of their actions and, as such, to better connect causes and consequences during social interactions. This is thought to be key for them to progressively understand that specific behaviors (words, gestures, etc.) are referential (i.e., they are caused by specific mental representations possessed by senders), which can then support a progressive internalization of their meanings [113,114,123].

In summary, the growing literature suggests that social contingency is a key factor for infant learning, including language acquisition, but the extent to which it also shapes the way infants attend to and process a speaker’s face remains unclear. A necessary next step for research on the use of AV speech cues for infants’ language learning will, thus, be, in our opinion, to examine how social contingency and other factors related to the dialogic nature of early caregiver–infant communication (e.g., turn-taking rates and structure, the predictability of social interactions, etc. [124,125]), shape infants’ disposition to selectively attend to and process the speakers’ faces, eyes and mouths.

### 4.2. The Influence of Speakers’ Characteristics

Another variable that typically differs from screen-based to real-life studies, and which has also been largely understudied, is the type of speakers that infants are presented with or interact with. Screen-based settings that measure infant attention to talking faces systematically use unrelated strangers that often talk in IDS. By contrast, in real-life settings, infants are typically interacting with their caregivers, who sometimes show less consistent uses of IDS, for instance, when infants are not responsive to their solicitations (e.g., [126]). It is, therefore, possible that a speaker’s familiarity and differential use of IDS could account for the differences in attention to faces observed between screen-based and real-life setting studies (see Section 3).

First, low familiarity with an unknown speaker could enhance infants’ attention to the speaker’s face and talking mouth: such a strategy could help infants build speech representations across speakers (i.e., speaker normalization): a necessary and non-trivial step in early language acquisition [127,128]. Paying more attention to the caregiver’s face could also help infants evaluate the expertise of the unfamiliar speaker, estimating, for instance, their social group. Indeed, categorizing the speaker as either belonging or not to the same social group as the infant (in or out group) can change the way they expect and process speech information from this person (see [129,130]).

Second, the more systematic use of IDS in screen-based than in real-life settings could also influence the amount of time infants spend looking toward the face and, more specifically, the talking mouth of their caregivers. Indeed, screen-based settings that experimentally manipulate this factor have shown that IDS is more often associated with faces [131,132] and elicits more mouth looks than adult-directed speech [75,80,81]. In other words, the salient acoustic features of IDS could help infants orient and sustain not only their auditory attention to the speech signal (see [133] for a compatible hypothesis) but also their visual attention to the face, and especially to the talking mouth of their caregivers. In line with this idea, caregivers have been shown to flexibly modify the pitch of their IDS as a function of their infant’s responsiveness/feedback [126]. Further research should study more precisely the acoustic features that might attract infants’ visual attention toward the face or mouth of speakers and whether this behavior is also observed in more complex and usually noisier real-life situations.

### 4.3. The Influence of Dyadic Bidirectional Contingency

The last variable that we believe deserves attention, and that also differs from screen-based to real-life studies, is the type of interaction or situation that infants are presented with during an experiment. Most of the evidence gathered from screen-based studies originates from one type of situation: a video showing a close-up of a speaker’s face, talking to the infant, against a neutral background, with no or a very small number of objects or other types of distractors. This situation is supposed to mimic everyday face-to-face interactions where parents talk to their infant while sitting on a chair (e.g., during eating, spoon-feeding) or laying down on his/her back (e.g., during a diaper change or before sleeping). During the first months of their life, while mobility is reduced, real dyadic interactions can resemble these—where faces are at a short distance—and observations of caregivers and their infants younger than 9–12 months reflect this [24,25]. However, once infants are older and start to determine their visual inputs mostly by themselves, their experiences become more complex (see Section 2.1), and, therefore, screen-based studies fail to capture infants’ most usual types of interaction. On the other hand, naturalistic studies have typically explored how infants deploy their attention in situations of free play with various toys, often in the context of a situation where caregivers and their infants sit opposite to each other around a table, hence diminishing the relative size and saliency of faces in comparison to most screen-based settings [24,25].

These situational differences are relevant and should be considered when drawing conclusions about how infants deploy their attention, as infants’ attentional strategies are known to be task-dependent [100,134,135], and, therefore, infants adapt and change their behavior according to the typology of these situations. Prior studies using head-mounted eye trackers have shown that, indeed, infants mostly modulate their attention to the body and face of their mother as a function of task/situation (e.g., reaching, removing obstacles, crawling) and the mother’s position and actions [136].

Further research should thus explore the dynamics of infants’ selective attention across various situations (e.g., face-to-face interaction during feeding or table-top play sessions vs. side-by-side play interactions on a mat, etc.) in order to understand the way in which these specific settings (i.e., situational or task demands) modulate infants’ reliance on the AV speech cues of talking faces.

## 5. Conclusions and Directions for Future Research

In the present paper, we discuss the role of infants’ attention to talking faces as a mechanism to enhance speech processing and language learning (see also [39,40] for screen-based studies reviews). The originality of our approach is that it puts into perspective findings from both screen-based and real-life settings and then focuses on the specific factors that might be most influential yet are currently understudied in both methodologies that explore AV speech perception for early language acquisition.

While it appears clear that attending and processing AV speech cues from talkers’ faces is an important mechanism that supports infants’ language learning, we have seen that studies to date remain quite specific to their context and methodology (e.g., screen-based, non-contingent paradigms, or real-life, observational-only paradigms, all providing mostly correlational evidence) and that there are several important factors that need to be better explored in order to reduce the gap between our experimental studies and infants’ real-life learning mechanisms.

Crucially, given that infants’ attention to faces in real-life interactions is less frequent than previously thought or observed by screen-based studies and that it diminishes with increasing age, it becomes highly relevant for future studies to identify in which specific situations of their daily life, such as from the perspective of language learning, infants and children seek AV speech information in their caretaker’s faces. To approach this question, we might have to explore factors pertaining to the specific task or experimental situation, resembling real-life routines (e.g., play, book sharing, eating) that are known to foster language acquisition, but also factors relative to the person addressing the infant (e.g., caregiver, unknown experimenter, foreign language speaker, etc.). Finally, we believe that the dynamic aspects of communication, such as social contingency (i.e., the adaptation of infants’ behavior to the caregiver’s face and vice versa), should also be explored in future studies, aiming to move away from “passive-learning” paradigms and introduce contingency and infants’ intrinsic motivation to move closer to real-life learning situations.

Given that infants allocate their attention as a function of their current informational needs and goals [137,138,139], it is likely that the above-mentioned factors (i.e., speakers’ familiarity, task dependencies, and social contingency) change infants’ motivation to learn from and about speakers [140], thus modulating their attention toward talking faces, and, in turn, learning. Future studies that systematically vary the relevance of speakers, faces, and contrasting stimuli using different methodologies could help to better understand the extent to which infants’ early preference for faces is perceptual and category-specific rather than attention-based and dependent on self-relevance and motivational factors.

Finally, this perspective also suggests that inter-individual differences in infants’ typical home environment and caregiving practices could greatly modulate infants’ attention and reliance on talking faces for learning their native language, for instance, as a function of the relative frequency of face-to-face vs. side-to-side interactions in their homes. Most research in this area has focused on Westerners, so it remains plausible that exploiting visual information from talking faces is only one of many strategies that infants could use to support language acquisition. For instance, face-to-face interactions involving a mutual gaze are thought to be more common in Western cultures compared to side-to-side interactions involving mutual touch in other cultures [141]. It is, therefore, possible that other strategies and modalities (e.g., touch, gestures) are more privileged than the visual modality outside of Western cultural settings. Studying infants’ attention to talking faces and bodies in culturally more varied samples and real-life interactions is, therefore, an important venue for future research and is important to better document diversities and universalities in the ways infants learn their native language/s.

## Data Availability

Not applicable.

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
