# Peer review of "The Role of Talking Faces in Infant Language Learning: Mind the Gap between Screen-Based Settings and Real-Life Communicative Interactions"

_brainsci, 2023, doi:10.3390/brainsci13081167_

Round 1
Reviewer 1 Report
The paper is not presenting new experimental results since it is a systematic review. However, it is a very well-done, well-conceived and well-written one to deserve to be published. I do believe that the scientific community may benefit from the work the authors have done in considering all of the most relevant experimental evidence accrued so far, and in highlighting some relevant issues that are not always taken into consideration in experiment design.
Baby-direct speech with emphasis on the mouth area is suggested as a way to boost phoneme detection and lexical acquisition in children’s mother language.
I have only one very local comment:
- line 144: from and 8 months… I suppose something is missing.
Author Response
Dear reviewer,
Thank you very much for revising our article and for your constructive feedback. We have addressed the comments and corrected the errors indicated by you and reviewer 2, and we have marked in red the corrections / additions. We hope that you may find the revised version acceptable for publication.
Sincerely,
Joan Birulés

Reviewer 2 Report
The study "The role of talking faces in infant language learning: Mind the gap between screen-based settings and real-life communicative interactions" is a thorough review. In the introduction and part 2 of the paper, the authors naturally introduce the reader to the theoretical framework, and existing knowledge and focus on the factors that are important to the study.
The purpose and how the literature review will be approached are adequately described. The subject is approached globally and diversely, covering different extensions for visual and audio processing.
Some of the suggestions to improve the work initially concern field two, where there are some points where the citation of the bibliography should be corrected so that it follows the standard of the journal and is harmonized with the rest of the text (e.g., in line 90, 118, 131,136,177,190,192,195). After making the corresponding adjustments and corrections recommended by the reviewers, please check, and change the numbering where necessary.
The connection of attention and language in field three is done very carefully and guides the reader by framing the issue nicely with completeness. The addition of studies with good documentation and reference to EBP is particularly important as it adds more value to the study results and increases the scope of scientific interest.
I also think the first paragraph in field 4 needs to be strengthened with a bibliography and an apparent reference to the bibliography, which is missing in this version.
There is also a typo in the title 4.2.
Field 4 correctly completes the factors concerning the dyadic level of interaction. This information offers essential clues to understanding the subject.
Finally, I would like to refer to the conclusions and future implications of the study. The study's findings are presented comprehensively and clearly, and the future extensions are well documented. The point emphasized at the end by the authors about the importance of the mother tongue, I believe, offers a lot to the conclusions
Author Response
Dear reviewer,
Thank you very much for revising our article and for your constructive feedback. We have addressed the comments and corrected the errors indicated by you and by reviewer 1, and we have marked in red the corrections / additions.
These include the correction of citations and their numbering, the modification of paragraph 4 (we have added several citations and a sentence to strengthen the paragraph, as suggested), and we have corrected the typo in the title of 4.2.
We hope that you may find the revised version acceptable for publication.
Look forward to your response,
Sincerely,
Joan Birulés
